# Generalization Analysis on Learning with a Concurrent Verifier

**Masaaki Nishino,    Kengo Nakamura,    Norihito Yasuda**
NTT Communication Science Laboratories, NTT Corporation
{masaaki.nishino.uh, kengo.nakamura.dx, norihito.yasuda.hn}@hco.ntt.co.jp

## Abstract

Machine learning technologies have been used in a wide range of practical systems. In practical situations, it is natural to expect the input-output pairs of a machine learning model to satisfy some requirements. However, it is difficult to obtain a model that satisfies requirements by just learning from examples. A simple solution is to add a module that checks whether the input-output pairs meet the requirements and then modifies the model's outputs. Such a module, which we call a *concurrent verifier* (CV), can give a certification, although how the generalizability of the machine learning model changes using a CV is unclear. This paper gives a generalization analysis of learning with a CV. We analyze how the learnability of a machine learning model changes with a CV and show a condition where we can obtain a guaranteed hypothesis using a verifier only in the inference time. We also show that typical error bounds based on Rademacher complexity will be no larger than that of the original model when using a CV in multi-class classification and structured prediction settings.

## 1   Introduction

As machine learning technology matures, many systems have been developed that exploit machine learning models. When developing a system that uses a machine learning model, a model with merely small prediction error is not satisfactory due to real-field requirements. For example, an object recognition model that is sensitive to slight noise would cause security issues [4, 26], or a model with unexpected output would increase a system's cost for dealing with it. Thus, we want the input-output pairs of a machine learning model to satisfy some *requirements*. However, it is difficult to obtain a model that satisfies the requirements by just learning from examples. Moreover, since the learned models tend to be complex and the input domain tends to be quite large, it is unrealistic to certify that every input-output pair satisfies the requirements. In addition, even if we find an input-output pair that does not satisfy the requirements, modifying a model is difficult since we have to re-estimate it from the training examples.

This paper considers a way to obtain a machine learning model whose input-output pairs satisfy the required properties. We address the following assumptions for a situation where a machine learning model is used. First, we can judge whether input-output pair $(x, h(x))$ satisfies the requirements, where $h : \mathcal{X} \to \mathcal{Y}$ is a machine learning model or a hypothesis. As we show below, important use cases fit this setting. Second, a machine learning model already exists whose prediction error is small enough, although its input-output pairs are not guaranteed to satisfy the requirements. This second assumption is also reasonable since modern machine learning models show sufficient prediction accuracy in various tasks. Under these assumptions, a practical choice for addressing this problem isn't changing the machine learning model but adding a module that checks the input-output pairs of machine learning model $h$. We call this module *a concurrent verifier* (CV). Fig. 1 shows the system configuration of a machine learning model with a CV. The verifier checks whether the input-output

36th Conference on Neural Information Processing Systems (NeurIPS 2022).

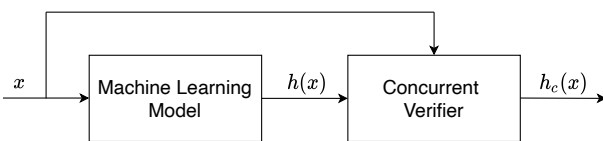

Figure 1: Overview of a machine learning model with a concurrent verifier that checks whether input-output pairs of a model satisfy requirements.

pair $(x, h(x))$ satisfies the required properties. If it satisfies the requirements, it outputs $h(x)$. If not, then it rejects $h(x)$ and modifies or requests the learning model to modify its output. A machine learning model and verifier pair can be seen as another machine learning model whose input-output pairs are guaranteed to satisfy the required conditions.

Although a model with a verifier can guarantee that its input-output pairs satisfy requirements, its effect on prediction error is unclear. This paper gives theoretical analyses of the generalization errors of a machine learning model with a CV. We focus on how the learnability of the original model, denoted as hypothesis class $\mathcal{H}$, can change by using the verifier. First, we consider a situation where we use a CV only in the inference phase. This setting corresponds to a case where the required properties are unknown when we are in the training phase. If the hypothesis class is PAC-learnable, we can obtain a guaranteed hypothesis using a verifier only in the inference time.

Second, we consider a situation where we know the requirements when learning the model. This situation corresponds to viewing the learnability of hypothesis set $\mathcal{H}_c$, which is obtained by modifying every hypothesis $h \in \mathcal{H}$ to satisfy the requirements. Hence we compare the generalization error upper bounds of $\mathcal{H}_c$ with those of $\mathcal{H}$. On the multi-class classification setting, we show that existing error bounds [13, 16] based on the Rademacher complexity of $\mathcal{H}$ are also bounds of modified hypothesis $\mathcal{H}_c$ for any input-output requirements. Moreover, we give similar analyses for a structured prediction task, which is a kind of multi-class classification where set of classes $\mathcal{Y}$ can be decomposed into substructures. It is worth analyzing the task since many works address the constraints in structured prediction. Some works give error bounds for structured prediction tasks, which are tighter than simply applying the bound for multi-class classification tasks [14, 6, 17]. Similar to the case of multi-class classification, we show that existing Rademacher complexity-based bounds for the structured prediction of $\mathcal{H}$ are also the bounds for $\mathcal{H}_c$.

Our main contributions are as follows: a) We introduce a concurrent verifier, which is a model-agnostic way to guarantee that machine learning models satisfy the required properties. Although a similar mechanism was used in some existing models, our model gives a generalization analysis that does not depend on a specific model. b) We show that if hypothesis class $\mathcal{H}$ is PAC-learnable, then using a verifier at the inference time can give a hypothesis with a guarantee in its generalization error. Interestingly, if H is not PAC-learnable, we might fail to obtain a guaranteed hypothesis even if the requirements are consistent with distribution $\mathcal{D}$. c) We show that if we use a CV in a learning phase of multi-class classification tasks, then the theoretical error bounds of $\mathcal{H}$ based on the Rademacher complexity will not increase with any input-output requirements. We also give similar results for structured prediction tasks.

## 1.1 Use Cases of a Concurrent Verifier

The following are some typical use cases for CVs.

**Error-sensitive applications:** A typical situation where we want to use a verifier is that some prediction errors might cause severe effects, which we want to avoid. For example, a recommender system might limit the set of candidate items depending on user attributes. Although such a rule might degrade the prediction accuracy, practically a safer model is preferable.

**Controlling outputs of structured prediction:** Constraints are frequently used in structured prediction tasks for improving the performance or the controllability of the outputs. For example, some works [21, 5] exploited the constraints on sequence labeling tasks for reflecting background knowledge to improve the prediction results. More recently, some works [9, 2] exploited the constraints in language generation tasks, including image captioning and machine translation, and restricted a

model to output a sentence that includes given keywords. Since the constraints used in this previous work can be written as a logical formula, our CV model can represent them as requirements.

**Robustness against input perturbations:** If a machine learning model changes its output because we modified its input from $x$ to $x'$, which is very close to $x$, then the model is described as sensitive against a small change [25]. It might be a security risk if a model is sensitive since its behavior is unpredictable. Therefore, some methods evaluate and verify the robustness of neural networks against small perturbations [26, 4]. Existing verification methods check a machine learning model's robustness around input $x$ by determining whether $x'$ exists that is close to $x$ and whether model $f$ gives different outputs, i.e., $h(x) \neq h(x')$, for verification samples $x_1, \ldots, x_n$. Although these verification methods can test a model, they do not directly show how to obtain a robust model.

A CV can fix a model to achieve robustness around samples $x_1, \ldots, x_n$ by setting a rule of form: "$h(x')$ must equal $h(x_i)$ if $x'$ is close to $x_i$." Although this solution might not guarantee robustness where samples are scarce, adding enough non-labeled verification samples is often a reasonable choice.

## 2   Related Work

Machine learning models that can exploit constraints have been investigated in many research fields, including statistical symbolic learning and structured prediction. For example, Markov logic networks [20], Problogs [8], and probabilistic circuit models [11] integrate statistical models with symbolic logic formulations. Since these models can incorporate hard constraints represented by symbolic logic, they can guarantee input-output pairs. However, previous research focused on their practical performance and gave little theoretical analysis of their learnability when hard constraints are used. Moreover, previous works integrated the ability to exploit constraints into specific models. In contrast, our CV is model-agnostic and can be used in combination with a wide range of machine learning models.

Recently, the verification of machine learning models has been gathering more attention. Attempts have verified whether a machine learning model has the desired properties [4, 26, 10, 24]. Exact verification methods use integer programming (MIP) [26], constraint satisfaction (SAT) [18], and a satisfiable module theory (SMT) solver [10] to assess the robustness of a neural network model against input noise. These approaches aim to obtain models that fulfill the required properties. However, verification methods cannot help modify the models if they do not satisfy the requirements. If we want ML models to meet requirements, post-processing is needed as our concurrent verification model.

Other methods can give upper bounds on generalization error, including VC-dimension [27] and its extensions [7, 19], Rademacher complexity [3, 12], stability [23], and PAC-Bayes [15, 1]. We use Rademacher complexity in the following analysis since it is among the most popular tools for giving theoretical upper bounds on generalization error. Rademacher complexity also has some extensions, including local Rademacher complexity [13] and factor graph Rademacher complexity [6]. We can provide theoretical guarantees on these extended measures.

## 3   Preliminaries

Our notation follows a previous work [22]. We first introduce the notations used in the following sections. Let $\mathcal{X}$ denote the domain of the inputs, let $\mathcal{Y}$ be the domain of the labels, and let $Z$ be the domain of the examples defined as $Z := \mathcal{X} \times \mathcal{Y}$. Let $\mathcal{H}$ be a hypothesis class, and let $\ell : \mathcal{H} \times Z \to \mathbb{R}_+$ be a loss function. Training data $S = (z_1, \ldots, z_m) \in Z^m$ is a finite sequence of size $m$ drawn i.i.d. from a fixed but unknown probability distribution $\mathcal{D}$ on $Z$. Learning algorithm $A$ maps training data $S$ to hypothesis $h$. We use notation $A(S)$ to denote the hypothesis that learning algorithm $A$ returns upon receiving $S$. We represent set $\{1, \ldots, K\}$ as $[K]$.

Given distribution $\mathcal{D}$ on $Z$, we denote by $L_{\mathcal{D}}(h)$ *the generalization error* and by $L_S(h)$ *the empirical error* of $h$ over $S$, defined by

$$L_{\mathcal{D}}(h) := \mathop{\mathbb{E}}_{z \sim \mathcal{D}} [\ell(h, z)], \quad L_S(h) := \frac{1}{m} \sum_{i=1}^{m} \ell(h, z_i). \tag{1}$$

**PAC learnability:** We introduce PAC learnability and agnostic PAC learnability as follows.

**Definition 3.1.** (Agnostic PAC learnability) Hypothesis class $\mathcal{H}$ is *agnostic PAC-learnable* if there exists function $m_{\mathcal{H}} : (0,1)^2 \to \mathbb{N}$ and learning algorithm $A$ with the following property: For every $\epsilon, \delta \in (0,1)$ and distribution $\mathcal{D}$ over $Z$, if $S$ consists of $m \geq m_{\mathcal{H}}(\epsilon, \delta)$ i.i.d. examples generated by $\mathcal{D}$, then with at least probability $1 - \delta$, the following holds:

$$L_{\mathcal{D}}(A(S)) \leq \min_{h' \in \mathcal{H}} L_{\mathcal{D}}(h') + \epsilon. \tag{2}$$

Distribution $\mathcal{D}$ is *realizable* by hypothesis set $\mathcal{H}$ if $h^* \in \mathcal{H}$ exists such that $L_{\mathcal{D}}(h^*) = 0$. If $\mathcal{D}$ is realizable by agnostic PAC-learnable hypothesis $\mathcal{H}$, then $\mathcal{H}$ is *PAC-learnable*. If $\mathcal{H}$ is PAC-learnable, then Eq. (2) becomes $L_{\mathcal{D}}(A(S)) \leq \epsilon$ since $\min_{h' \in \mathcal{H}} L_{\mathcal{D}}(h') = 0$.

**Rademacher complexity:** In the following sections, we use Rademacher complexity for deriving the generalization bounds. Given loss function $\ell(h, z)$ and hypothesis class $\mathcal{H}$, we denote $\mathcal{G}$ as

$$\mathcal{G} := \ell \circ \mathcal{H} := \{z \mapsto \ell(h, z) : h \in \mathcal{H}\}.$$

**Definition 3.2.** (Empirical Rademacher complexity) Let $\mathcal{G}$ be a family of functions mapping from $Z$ to $\mathbb{R}$, and let $S = (z_1, \ldots, z_m) \in Z^m$ be the training data of size $m$. Then *the empirical Rademacher complexity* of $\mathcal{G}$ with respect to $S$ is defined:

$$R_S(\mathcal{G}) := \underset{\boldsymbol{\sigma}}{\mathbb{E}} \left[ \sup_{g \in \mathcal{G}} \sum_{i=1}^{m} \sigma_i g(z_i) \right],$$

where $\boldsymbol{\sigma} = (\sigma_1, \ldots, \sigma_m) \in \{\pm 1\}^m$ are random variables distributed i.i.d. according to $\mathbb{P}[\sigma_i = 1] = \mathbb{P}[\sigma_i = -1] = 1/2$. *The Rademacher complexity* of $\mathcal{G}$ is defined as the expected value of the empirical Rademacher complexity:

$$R_m(\mathcal{G}) := \underset{S \sim \mathcal{D}^m}{\mathbb{E}} [R_S(\mathcal{G})].$$

# 4 Concurrent Verifier

Next we give a formal definition of a CV. A CV works with a machine learning model, which is function $h : \mathcal{X} \to \mathcal{Y}$. If $x$ is given to the model, which outputs $h(x)$, then the verifier checks whether $(x, h(x))$ satisfies the required property. We assume that the required property can be represented as *requirement function* $c : (\mathcal{X} \times \mathcal{Y}) \to \{0, 1\}$. If $c(x, h(x)) = 1$, then the pair satisfies the property; if $c(x, h(x)) = 0$, then it does not. Requirement function $c$ can be represented by a set of deterministic rules. For example, if $\mathcal{X} = \mathbb{R}$ and $\mathcal{Y} = \{0, 1\}$, then the requirements can be in the following form: "if $x > 0$, then $y \neq 0$." We assume that for all possible input $x \in \mathcal{X}$, there exists $y \in \mathcal{Y}$ such that $c(x, y) = 1$ for avoiding the situation where the requirements are unsatisfiable for any output $y$. This assumption can be easily relaxed if we allow a machine learning model to reject unsatisfiable input $x$.

After checking the input-output pair, a verifier modifies output $h(x)$ depending on the value of $c(x, h(x))$. If $c(x, h(x)) = 1$, the verifier outputs $h(x)$ since it satisfies the requirements. If $c(x, h(x)) = 0$, then the verifier modifies $h(x)$ to some $y \in \mathcal{Y}$ that satisfies $c(x, y) = 1$. If we use a verifier with a machine learning model that corresponds to $h$, then the combination of the model and the verifier can be seen as function $h_c : \mathcal{X} \to \mathcal{Y}$, defined as

$$h_c(x) := \begin{cases} h(x) & \text{if } c(x, h(x)) = 1 \\ y_c & \text{if } c(x, h(x)) = 0 \end{cases}, \tag{3}$$

where $y_c \in \mathcal{Y}$ satisfies $c(x, y_c) = 1$ and is selected deterministically. When $\mathcal{Y} = [K]$, an example for selecting minimum $i \in [K]$ satisfying $c(x, i) = 1$ as $y_c$ is a reasonable choice. When $\mathcal{Y} = [K]$ and $h(x)$ is made by scoring functions $h(x, y) : (\mathcal{X} \times \mathcal{Y}) \to \mathbb{R}$, it is also reasonable to select $y^*$ such that $y^* = \operatorname{argmax}_{y \in \mathcal{Y}, c(x,y)=1} h(x, y)$. Learning a model corresponds to selecting hypothesis $h$ from hypothesis class $\mathcal{H}$. Therefore, learning a model with a CV corresponds to choosing a hypothesis from the modified hypothesis class: $\mathcal{H}_c = \{h_c : h \in \mathcal{H}\}$. By definition, every hypothesis in $\mathcal{H}_c$ satisfies the requirements, and thus we can guarantee that the model satisfies the condition if we select a hypothesis from $\mathcal{H}_c$. In the following sections, we analyze the learnability of $\mathcal{H}_c$ by comparing it with that of $\mathcal{H}$.

## 5 Inference Time Verification

We first analyze the change of the generalization errors when we use a verifier only in an inference phase. In other words, requirements are unknown in the learning phase, and we estimate hypothesis $\hat{h} = A(S)$ from hypothesis class $\mathcal{H}$ by using training data $S$ and algorithm $A$. In the inference phase, we use a CV to modify $\hat{h}$ to $\hat{h}_c$. We call this setting the *inference time verification* (ITV). This class of situations contains many exciting settings: 1) pre-trained machine learning models used in a wide range of applications, and 2) models that are hard to replace, which might encounter different requirements from those at the learning time in the long run.

In this section, we give analyses on a multi-class classification setting. We set $\mathcal{Y} = [K]$, and hypothesis class $\mathcal{H}$ is set of mappings $h : \mathcal{X} \to [K]$. We also assume that loss function $\ell$ is 0-1 loss defined as $\ell_{\text{0-1}}(h, (x, y)) \coloneqq \mathbf{1}_{h(x) \neq y}$, where $\mathbf{1}$ is an indicator function.

The following theorem shows a situation where ITV works well: a situation where the generalization error of $\hat{h}_c$ does not exceed that of the other hypotheses in $\mathcal{H}_c$ with high probability.

**Theorem 5.1.** *If $\mathcal{Y} = [K]$, and hypothesis class $\mathcal{H}$ is PAC-learnable with 0-1 loss $\ell_{\text{0-1}}$, training data $S$, and algorithm $A$, then suppose that $\hat{h} = A(S)$ is a hypothesis estimated form $S$ satisfying $L_{\mathcal{D}}(\hat{h}) \leq \epsilon$ for some parameter $\epsilon \in (0, 1)$. Then for any requirement c, hypothesis $\hat{h}_c$ obtained by modifying $\hat{h}$ with a CV satisfies*

$$L_{\mathcal{D}}(\hat{h}_c) \leq \min_{h_c \in \mathcal{H}_c} L_{\mathcal{D}}(h_c) + \epsilon.$$

We give a proof in Appendix A. The proof bounds $L_{\mathcal{D}}(\hat{h}_c)$ using the fact that it is close to $L_{\mathcal{D}}(f_c)$, where $f_c$ is obtained by modifying $f : \mathcal{X} \to \mathcal{Y}$ to satisfy $L_{\mathcal{D}}(f) = 0$. The theorem suggests that if $\mathcal{H}$ is PAC-learnable, then inference time verification is sufficient to obtain a hypothesis with small generalization error in $\mathcal{H}_c$.

Note that the generalization error might increase with a verifier, and the amount of the increase is always larger than $L_{\mathcal{D}}(f_c)$. Therefore, $L_{\mathcal{D}}(f_c)$ represents the discrepancy between data distribution $\mathcal{D}$ and requirement $c$, which is *consistent* with $f$ if $c(x, f(x)) = 1$ for all $x$. If $c$ is consistent with $f$, then $L_{\mathcal{D}}(f_c) = 0$, and we can certify that $L_{\mathcal{D}}(\hat{h}_c) \leq \epsilon$.

The above theorem shows that ITV works when $\mathcal{H}$ is PAC-learnable. However, this will not hold if $\mathcal{D}$ is not realizable with $\mathcal{H}$, i.e, $\mathcal{H}$ is not PAC-learnable.

**Theorem 5.2.** *If $\mathcal{Y} = [K]$, the loss function is 0-1 loss $\ell_{\text{0-1}}$ and hypothesis class $\mathcal{H}$ is not realizable with $\mathcal{D}$, and then there exists training data $S$, algorithm $A$, requirements $c$, and $\epsilon \in (0, 1)$ such that $\hat{h} = A(S)$ satisfies $L_{\mathcal{D}}(\hat{h}) \leq \min_{h \in \mathcal{H}} L_{\mathcal{D}}(h) + \varepsilon$ but $L_{\mathcal{D}}(\hat{h}_c) > \min_{h_c \in \mathcal{H}_c} L_{\mathcal{D}}(h_c) + \epsilon$.*

We give in Appendix B a proof that shows a counterexample even if $c$ is consistent with ground truth $f$. The above theorems show that the realizability of $\mathcal{H}$ is the key factor that distinguishes among the cases where ITV works well. Moreover, unlike the realizable case, Theorem 5.2 holds even if requirement $c$ is consistent with distribution $\mathcal{D}$. Let $f_{\mathcal{D}} : \mathcal{X} \to \mathcal{Y}$ be defined as the Bayes optimal predictor:

$$f_{\mathcal{D}}(x) \coloneqq \underset{y \in \mathcal{Y}}{\operatorname{argmax}} \, \mathbb{P}[y \mid x].$$

The Bayes optimal predictor is optimal, in the sense that for every other classifier $g : \mathcal{X} \to \mathcal{Y}$, $L_{\mathcal{D}}(f_{\mathcal{D}}) \leq L_{\mathcal{D}}(g)$. Theorem 5.2 holds if $c$ is consistent with $f_{\mathcal{D}}$. These results show that existing methods [9, 2] using constraints only in the inference time might fail to select the best hypothesis.

**Running time analysis:** Using a CV increases the time needed for inference. Suppose that a verifier is an oracle that can answer the query about the value of $c(x, y)$. To achieve a previously shown modification procedure (3), we need at most $K$ queries.

## 6 Learning Time Verification

In Section 5, we show that if $\mathcal{H}$ is PAC-learnable with 0-1 loss, then modifying a hypothesis at the inference time is sufficient to obtain a hypothesis with the smallest generalization error while

satisfying the requirements. If $\mathcal{H}$ is not PAC-learnable, then the ITV scheme might fail to obtain a hypothesis with small generalization error. Here we show that the generalization error can be bounded when we use a CV in the learning phase. We call this setting *learning time verification* (LTV).

Since the LTV scheme corresponds to a learning task where the hypothesis class is $\mathcal{H}_c$, we analyze the learnability of $\mathcal{H}_c$ using the standard tools for generalization analyses. This paper provides analyses based on Rademacher complexity since its a widely used tools that can give tight bounds for both data-dependent and data-independent cases. Moreover, some previous work gives bounds of structured prediction tasks using Rademacher complexity. In the literature, constraints are actively used in structured prediction tasks, including language generation and sequence labeling. Therefore, analyzing the generalization error is important when using a CV on structured prediction tasks.

In the following, we first show the upper bounds of generation error based on the Rademacher complexity of $\mathcal{H}_c$ in a multi-class classification task (§6.1, 6.2) and a structured prediction setting (§6.3). Our main finding is that the upper bounds based on the Rademacher complexity of $\mathcal{H}_c$ are always less than or equal to those of $\mathcal{H}$. Therefore, adding a CV to a machine learning model will not degrade its learnability.

## 6.1 Multi-class Classification

We first give the Rademacher complexity-based error bounds on a multi-class classification task, i.e., $\mathcal{Y} = [K]$. In this section, we show that a standard upper bound [16] based on the Rademacher complexity of $\mathcal{H}$ can be used as an upper bound of $\mathcal{H}_c$ for any requirement $c$. In the next section, we show that a state-of-the-art error bound, based on local Rademacher complexity $\mathcal{H}$, can also be used as an upper bound of $\mathcal{H}_c$.

Following previous works, let $h : (\mathcal{X} \times \mathcal{Y}) \to \mathbb{R}$ be a scoring function, and define hypothesis class $\mathcal{H}$ as a set of scoring functions. A scoring function defines a mapping from $\mathcal{X}$ to $\mathcal{Y}$:

$$x \mapsto \operatorname*{argmax}_{y \in \mathcal{Y}} h(x, y).$$

Let $\rho_h(x, y)$ be *the margin of function* of $h$:

$$\rho_h(x, y) := h(x, y) - \max_{y' \neq y} h(x, y').$$

Hypothesis $h$ misclassifies the labeled example $(x, y)$ if $\rho_h(x, y) \leq 0$. Thus, by using a margin function, the 0-1 loss can be represented as $\ell_{\text{0-1}}(h, z) = \mathbf{1}_{\rho_h(x,y) \leq 0}$. Since 0-1 loss is hard to handle during learning, we use margin loss $\ell_\rho(h, (x, y)) = \Phi_\rho(\rho_h(x, y))$, where $\Phi_\rho(t)$ is defined as

$$\Phi_\rho(t) = \min(1, \max(0, 1 - t/\rho)).$$

Function $f : \mathbb{R} \to \mathbb{R}$ is said to be $\mu$-*Lipschitz* if $|f(t) - f(t')| \leq \mu|t - t'|$ for any $t, t' \in \mathbb{R}$. $\Phi_\rho$ is an $1/\rho$-Lipschitz function. The *empirical margin loss* of hypothesis $h$ is defined as

$$L_{S,\rho}(h) := \frac{1}{m} \sum_{i=1}^m \Phi_\rho(\rho_h(x_i, y_i)).$$

Identical to the case of ITV, introducing a CV to a machine learning model corresponds to modifying its corresponding hypothesis class $\mathcal{H}$ to hypothesis class $\mathcal{H}_c$ that is consistent with requirement $c$. If $h$ is a score function, then we define consistent function $h_c$:

$$h_c(x, y) = \begin{cases} h(x, y) & \text{if } c(x, y) = 1 \\ -M & \text{if } c(x, y) = 0 \end{cases}, \tag{4}$$

where $M$ is a positive constant satisfying $M > |\max_{(x,y) \in Z} h(x, y)|$. As described in Section 4, we assume that there exists $y \in \mathcal{Y}$ that satisfies $c(x, y) = 1$ for all $x \in \mathcal{X}$. Therefore, we can guarantee that $\rho_{h_c}(x, y) < 0$ if $c(x, y) = 0$.

The following are the main results of the general multi-class learning problem, which is based on the margin bound shown in Theorem 9.2 of Mohri et al. [16]. Our main finding is that the generalization error of any hypothesis, $h_c$, is bounded by the Rademacher complexity of hypothesis set $\mathcal{H}$, which suggests that if we have a tight bound for hypothesis class $\mathcal{H}$, then we can expect to find a good hypothesis from $\mathcal{H}_c$ under any requirements $c$.

**Theorem 6.1.** *Let $\mathcal{H} \subseteq \mathbb{R}^{\mathcal{X} \times \mathcal{Y}}$ be a hypothesis class with $\mathcal{Y} = [K]$, and let $c$ be a requirement. Fix $\rho > 0$. Then for any $\delta > 0$, with probability at least $1 - \delta$, the following bound holds for all $h_c \in \mathcal{H}_c$:*

$$L_{\mathcal{D}}(h_c) \leq L_{S,\rho}(h_c) + \frac{4K}{\rho} R_m(\Pi_1(\mathcal{H})) + \sqrt{\frac{\log \frac{1}{\delta}}{2m}} ,$$

*where $\Pi_1(\mathcal{H})$ is defined as*

$$\Pi_1(\mathcal{H}) := \{x \mapsto h(x, y) : y \in \mathcal{Y}, h \in \mathcal{H}\} .$$

We give a proof in Appendix C. We obtain the results by showing that the upper bounds of the Rademacher complexity of $\mathcal{H}_c$ are bounded by some upper bounds of the Rademacher complexity of $\mathcal{H}$. All the proofs of the theorems in this section use similar techniques. Parameter $\rho$ sets the margin value. Following a previously shown technique [16], we obtain a generalized bound that holds uniformly for all $\rho > 0$. The above theorem suggests that using a CV at a learning phase does not worsen the error bound for any requirement $c$. Intuitively, the theorem seems reasonable since requirements $c$ imposes a restriction on $\mathcal{H}$, and thus the complexity of $\mathcal{H}_c$ is not larger than $\mathcal{H}$. However, it is not so trivial since $\mathcal{H}_c \subseteq \mathcal{H}$ is not always true.

**Running time analysis:** We analyze the number of evaluations $c(x, y)$ required for learning with a CV. Let $S_1$ be a sub-sequence of training example $S$ such that $c(x_i, y_i) = 1$, and let $S_0$ be a sub-sequence such that $c(x_i, y_i) = 0$. If we use a 0-1 loss function, then the empirical loss of hypothesis $h_c$ is

$$L_S(h_c) = \frac{1}{m} \sum_{(x_i, y_i) \in S_1} \mathbf{1}_{h_c(x_i) \neq y_i} + \frac{|S_0|}{m} ,$$

since every $h_c$ misclassifies the examples in $S_0$. Therefore, we need at most $K|S_1| + |S_0|$ queries for the learning process. This is also true when we use a margin loss function. On the other hand, the problem of estimating the best hypothesis might be more difficult than the original problem depending on requirement $c$.

## 6.2   Tighter Bound Based on Local Rademacher Complexity

The bound for $\mathcal{H}_c$ shown in the previous section is relatively simple, and tighter bounds of $\mathcal{H}$ based on the Rademacher complexity have been developed in the literature. In this section, we show that the state-of-the-art error bound for $\mathcal{H}$ based on *the local Rademacher complexity* can be used as a bound for $\mathcal{H}_c$ for any requirements $c$.

**Definition 6.2.** Let $\mathcal{G}$ be a family of functions from $Z$ to $\mathbb{R}$, and let $S$ be training data of size m. Then for any $r > 0$, *the empirical local Rademacher complexity* of $\mathcal{G}$ is defined as

$$R_S(\mathcal{G}; r) = R_S \left( \{ag : a \in [0, 1], g \in \mathcal{G}, \mathbb{E}[(ag)^2] \leq r\} \right) .$$

Li et al. [13] showed a tighter generalization bound for a multi-class classification problem using the local Rademacher complexity when the hypothesis class is a $\ell_p$ norm hypothesis space with kernel $\kappa$, defined as

$$\mathcal{H}_{p,\kappa} := \{h = (\langle \mathbf{w}_1, \phi(x) \rangle, \dots, \langle \mathbf{w}_K, \phi(x) \rangle) : \|\mathbf{w}\|_{2,p} \leq 1, 1 \leq p \leq 2\} ,$$

where $h$ is represented as a vector valued function $(h_1, \dots, h_K)$ with $h_j(x) = h(x, j), \forall j = 1, \dots, K$, and $\kappa : \mathcal{X} \times \mathcal{X} \to \mathbb{R}$ is a Mercer kernel with associated feature map $\phi$, i.e., $\kappa(x, x') = \langle \phi(x), \phi(x') \rangle$. $\mathbf{w} = (\mathbf{w}_1, \dots, \mathbf{w}_K)$, and $\|\mathbf{w}\| = \left[ \sum_{i=1}^{K} \|\mathbf{w}\|_2^p \right]^{\frac{1}{p}}$ is the $\ell_{2,p}$-norm. For any $p \geq 1$, let $q$ be the dual exponent of $p$ satisfying $1/p + 1/q = 1$. Let $\Phi : \mathbb{R} \to \mathbb{R}$ be a loss function satisfying the following: 1) $\mathbf{1}_{t<0}(t) \leq \Phi(t)$ for all $t$; 2) $\Phi(t)$ is decreasing and has zero point $c_{\Phi}$; 3) $\Phi$ is $\zeta$-smooth, that is, $|\Phi'(t) - \Phi'(t')| \leq \zeta|t - t'|$.

Let $\mathcal{H}_{p,\kappa,c}$ be the hypothesis class obtained by modifying hypothesis $\mathcal{H}_{p,\kappa}$ to satisfy requirements $c$, and $\mathcal{L}_c := \{(x, y) \mapsto \Phi(\rho_{h_c}(x, y)) : h_c \in \mathcal{H}_{p,\kappa,c}\}$. The following theorem gives a bound of the local Rademacher complexity of $\mathcal{L}_c$.

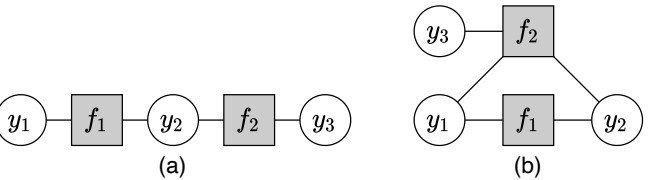

Figure 2: Example of factor graphs: (a) represents decomposition $h(x, y) = h_{f_1}(x, y_1, y_2) + h_{f_2}(x, y_2, y_3)$ and (b) represents decomposition $h(x, y) = h_{f_1}(x, y_1, y_2) + h_{f_2}(x, y_1, y_2, y_3)$.

**Theorem 6.3.** *Let $\mathcal{H}_{p,\kappa,c}$ be the set of hypotheses obtained by modifying hypothesis $h \in \mathcal{H}_{p,\kappa}$ with requirement $c$. For any $\delta > 0$, with probability at least $1 - \delta$, the following bound holds:*

$$R_m(\mathcal{L}_c; r) \leq \frac{C_{d,\vartheta}\xi(K)\sqrt{\zeta r}\log^{\frac{3}{2}}(m)}{\sqrt{m}} + \frac{4\log\frac{1}{\delta}}{m},$$

*where $\vartheta = \sup_{x \in \mathcal{X}} \kappa(x, x) < \infty$, $d = \sup_{t \in \mathbb{R}} \Phi(t) < \infty$, and $C_{d,\vartheta}$ is a constant. $\xi(K)$ is*

$$\xi(K) = \begin{cases} \sqrt{e}(4\log K)^{1+\frac{1}{2\log K}} & \text{if } q \geq 2\log K, \\ (2q)^{1+\frac{1}{q}}K^{\frac{1}{q}} & \text{otherwise .} \end{cases}$$

We give a proof in Appendix D. The bound equals that of $R_m(\mathcal{L}; r)$ (Theorem 1 of [13]) for any requirement $c$ and any hypothesis $\mathcal{H}_{\rho,\kappa}$. Therefore, the generalization error bounds based on Theorem 1 of Li et al. [13] holds for any requirement $c$.

### 6.3 Analyses of Structured Prediction

Structured prediction is a kind of multi-class classification task, where label set $\mathcal{Y}$ might be a set of sequences, images, graphs, trees, or other objects admitting some possibly overlapping structure. As mentioned in Section 1, previous works try to impose constraints on the output of structured prediction tasks. Thus it is also useful to derive error bounds for structured prediction tasks when we use a CV. In the following, we show that the Rademacher complexity-based generalization error bounds derived in a seminal work of Cortes et al. [6] also hold if we use a CV. Although tighter bounds are given in a more recent work [17, 14], we give bounds based on Cortes et al. [6] due to their simplicity.

We give some definitions for the structured prediction task. Following previous work, we assume that $\mathcal{Y}$ is decomposable along with substructures: $\mathcal{Y} = \mathcal{Y}_1 \times \cdots \times \mathcal{Y}_l$. Here $\mathcal{Y}_k$ is a set of possible labels that can be assigned to the $k$-th substructure. We denote by $\mathsf{L} : \mathcal{Y} \times \mathcal{Y} \to \mathbb{R}_+$ a loss function that measures the dissimilarity of two elements of output space $\mathcal{Y}$. $\mathsf{L}$ is *definite*, that is, $\mathsf{L}(y, y') = 0$ iff $y = y'$. A typical definite loss function for a structured prediction task is the Hamming loss defined by $\mathsf{L}(y, y') = \frac{1}{l} \sum_{k=1}^{l} \mathbf{1}_{y_k \neq y'_k}$ for all $y = (y_1, \ldots, y_l)$ and $y' = (y'_1, \ldots, y'_l)$, with $y_k, y'_k \in \mathcal{Y}_k$. Other typical examples of loss functions can be seen in Cortes et al. [6]. Using loss function $\mathsf{L}$, the generalization and empirical error of $h$ are defined:

$$L_{\mathcal{D}}(h) = \mathbb{E}_{(x,y) \sim \mathcal{D}}[\mathsf{L}(\mathsf{h}(x), y)], \quad L_S(h) = \frac{1}{m} \sum_{i=1}^{m} \mathsf{L}(\mathsf{h}(x), y).$$

As with the multi-class classification task, hypothesis class $\mathcal{H}$ can be represented as a set of scoring function $h : \mathcal{X} \times \mathcal{Y} \to \mathbb{R}$. We use $\mathsf{h}(x)$ to represent the predictor defined by $h \in \mathcal{H}$: $\mathsf{h}(x) := \operatorname{argmax}_{y \in \mathcal{Y}} h(x, y)$ for all $x \in X$. Following the previous work, we assume that each scoring function can be decomposed as a sum, and such decomposition follows a *factor graph*. Factor graph $G$ is a tuple $G = (V, F, E)$, where $V$ is a set of variable nodes, $F$ is a set of factor nodes, and $E$ is a set of undirected edges between a variable node and a factor node. Every node in $V$ corresponds to a substructure index, where $V = \{1, \ldots, l\}$.

For any factor node $f$, we denote by $\mathcal{N}(f) \subseteq V$ a set of variable nodes connected to $f$ and define $\mathcal{Y}_f$ as substructure set cross-product $\mathcal{Y}_f = \prod_{k \in \mathcal{N}(f)} \mathcal{Y}_k$. Then $h$ admits the following decomposition as

a sum of functions $h_f$, each taking as an argument a pair of $(x, y_f) \in \mathcal{X} \times \mathcal{Y}_f$:

$$h(x, y) = \sum_{f \in F} h_f(x, y_f). \tag{5}$$

Figure 2 shows examples of decompositions based on factor graphs. We conventionally assume that the structure of the factor graphs may change depending on a particular example $(x_i, y_i)$: $G(x_i, y_i) = G_i = ([l_i], F_i, E_i)$. A special case of this setting is when size $l_i$ of each example is allowed to vary. In such a case, the number of possible labels $\mathcal{Y}$ is potentially infinite.

Following multi-class classification, our CV maps hypothesis $h$ to $h_c$ to satisfy the requirements. The definition of $h_c$ follows Eq. (4). This definition does not require $h_c$ to have a factored representation.

For analyzing the complexity, Cortes et al. [6] introduced *empirical factor graph Rademacher complexity* $R_S^G(\mathcal{H})$ of hypothesis class $\mathcal{H}$ for $S = (x_1, \ldots, x_m)$ and factor graph $G$:

$$R_S^G(\mathcal{H}) = \frac{1}{m} \mathbb{E}_{\boldsymbol{\epsilon}} \left[ \sup_{h \in \mathcal{H}} \sum_{i=1}^{m} \sum_{f \in F_i} \sum_{y \in \mathcal{Y}_f} \sqrt{|F_i|} \epsilon_{i,f,y} h_f(x_i, y) \right],$$

where $\boldsymbol{\epsilon} = (\epsilon_{i,f,y})_{i \in [m], f \in F_i, y \in \mathcal{Y}_f}$ and every $\epsilon_{i,f,y}$ is i.i.d. a Rademacher random variable. Factor graph Rademacher complexity of $\mathcal{H}$ for factor graph $G$ is defined as expectation

$$R_m^G(\mathcal{H}) := \mathbb{E}_{S \sim \mathcal{D}^m} [R_S^G(\mathcal{H})].$$

By using the factor graph Rademacher complexity, Cortes et al. [6] gives bounds for a structured prediction task with the following additive and multiplicative empirical losses:

$$L_{S,\rho}^{\mathrm{add}}(h) := \frac{1}{m} \sum_{i=1}^{m} \left[ \Phi^* \left( \max_{y' \neq y_i} \mathsf{L}(y', y_i) - \frac{1}{\rho} (h(x_i, y_i) - h(x_i, y')) \right) \right]$$

$$L_{S,\rho}^{\mathrm{mult}}(h) := \frac{1}{m} \sum_{i=1}^{m} \left[ \Phi^* \left( \max_{y' \neq y_i} \mathsf{L}(y', y_i) \left( 1 - \frac{1}{\rho} (h(x_i, y_i) - h(x_i, y')) \right) \right) \right],$$

where $\Phi^*(t) = \min(B, \max(0, t))$ for all $t$, with $B = \max_{y, y'} \mathsf{L}(y, y')$. As shown in [6], these loss functions cover typical surrogate loss functions used in structured prediction tasks. We show the following bound for structured predictions.

**Theorem 6.4.** *Fix $\rho > 0$. For any $\delta > 0$ and requirement c, with probability at least $1 - \delta$ over the draw of sample S of size m from distribution $\mathcal{D}$, the following holds for all $h_c \in \mathcal{H}_c$:*

$$L_{\mathcal{D}}(h_c) \leq L_{\mathcal{D},\rho}^{\mathrm{add}}(h_c) \leq L_{S,\rho}^{\mathrm{add}}(h_c) + \frac{4\sqrt{2}}{\rho} R_m^G(\mathcal{H}) + B \sqrt{\frac{\log \frac{1}{\delta}}{2m}}$$

$$L_{\mathcal{D}}(h_c) \leq L_{\mathcal{D},\rho}^{\mathrm{mult}}(h_c) \leq L_{S,\rho}^{\mathrm{mult}}(h_c) + \frac{4\sqrt{2}B}{\rho} R_m^G(\mathcal{H}) + B \sqrt{\frac{\log \frac{1}{\delta}}{2m}}.$$

We give a proof in Appendix E. $\rho$ is a parameter that determines the margin. Similar to the case of multi-class classification, we can derive a bound that holds for any $\rho > 0$ following a previous derivation [6]. The above result indicates that the bound will not change if we use a CV for any requirement $c$. This is interesting since the above result holds even if we do not have a factored representation of $h_c(x, y)$, similar to Eq. (5), although the derived bound depends on the factor graph Rademacher complexity, which depends on the factored representation of $h(x, y)$.

We analyzed the overhead of the running time for evaluating loss function $L_{S,\rho}^{\mathrm{add}}(h_c)$ and $L_{S,\rho}^{\mathrm{mult}}(h_c)$ for hypothesis $h_c$. Different from the multi-class classification case, both the number of queries and the overhead of the running time for the loss evaluation when we use a CV depend on the model and the type of requirements for structured predictions. This result is consistent with the literature, which reports that for structured prediction tasks, original tractable optimization problems can be intractable if we put additional constraints [21].

# 7 Conclusion

This paper gives a generalization analysis when there are requirements that the input-output pairs of a machine learning model must satisfy. We introduce a concurrent verifier, a simple module that enables us to guarantee that the input-output pairs of a machine learning model satisfy the requirements. We show a situation where we can obtain a hypothesis with small error when we use a verifier only in the inference phase. Interestingly, if $\mathcal{H}$ is not PAC-learnable, we might fail to obtain a guaranteed hypothesis even if the requirements are consistent with distribution $\mathcal{D}$. We also give the generalization bounds based on Rademacher complexity when we use a verifier in a learning phase and find that the obtained bounds are less than or equal to the existing ones, independent of the machine learning model and the type of requirements.

## Acknowledgements

The authors thank the anonymous reviewers for their valuable feedback, corrections, and suggestions. This work was supported by JST PRESTO (Grant Number JPMJPR20C7, Japan), JST CREST (Grant Number JPMJCR22D3, Japan) and JSPS KAKENHI (Grant Number JP20H05963, Japan).

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
