# OpenReview forum: "Generalization Analysis on Learning with a Concurrent Verifier"
_NeurIPS.cc/2022/Conference — NeurIPS 2022 Accept_

### Official Review · Reviewer_TPUk · 2022-07-15

**Rating:** 7
**Confidence:** 2
**Soundness:** 3 good
**Presentation:** 3 good
**Contribution:** 3 good

**Summary:**

The paper analyzes the generalization bounds of ML models augmented with concurrent verifiers.
As background it presents PAC learnability and Rademacher complexity, and then it introduces concurrent verifiers.
It shows how generalization changes when a verifier is applied to a model at inference time when both when the hypothesis space is realizable and when it is not: when H is PAC learnable with 0-1 loss, inference time verification (ITV) gives the minimum possible generalization error while satisfying the constraints; when H is not PAC learnable, this is not obtained. The paper also considers learning time verification (LTV), using Rademacher complexity to show that adding a concurrent verifier to a model at learning time does not reduce its learnability, giving bounds both for multiclass prediction and structured prediction.


**Questions:**

* Do you have intuitive explanations for each of the results that could enhance the clarity of the paper?
* Are there practical settings where the Rademacher complexity based error bounds are known to be meaningful/not terribly loose, and also in which a verifier is useful?
* You show that Rademacher complexity based error bounds will be not larger than those of the original model when a CV is added in. How does this then allow you to conclude that the CV doesn't hurt the model's generalizability (line 15)? For example in modern ML systems my understanding is that typical generalization performances can be _well_ within these bounds, and hence preserving the bounds does not directly entail preserving generalization.


**Limitations:**

The conditions in the theorems capture the limitations of the results. e.g. for ITV the generalization guarantee requires that H is PAC learnable.

**Strengths And Weaknesses:**

### Originality

The results are original, drawing upon bounds derived in earlier work and extending them to the concurrent verifier settings.
The background material is both cited appropriately and explained clearly. The relevant bounds built upon are cited, and tighter bounds that are not built upon (but perhaps could be) are also referenced.

### Quality

All claims are well supported, with proofs provided in the appendices. I have only verified the proofs through section 5, not those of section 6. I have not found evidence of anything unsound in my review, and the results are sensible, so I expect even the proofs I have not verified thoroughly will be sound too.

### Clarity

The submission is well written and well organized. The background material is clearly explained. Most concepts are introduced prior to their use (at least up until line 280; e.g. Mercer kernels are not introduced).

The following are largely typographic suggestions, but these issues do not meaningfully degrade the overall quality of the paper.
* nit: Line 134 consists should read "consisting"
* nit: Line 156 "all" should be "allow"
* Line 185, it should say "sample" rather than "example", yes?
* nit: Line 185, s/be/is
* Line 191: The notation $f$ and $f_c$ is only introduced in the Appendix, and so seems out of scope here
* <strike>Line 237: if $\rho_h = 0$, is this necessarily a misclassification?</strike>
* Line 275: What does $f$ refer to in the definition of empirical local Rademacher complexity?

### Significance

The new setting, machine learning with a concurrent verifier, is an important setting for many application domains. I do not know whether the bounds are relevant/meaningful in these application domains in practice, as my understanding is that in modern ML systems typical generalization performances can be well within the bounds given by Rademacher complexity. Nevertheless, the setting is an important one and the results are (to the best of my knowledge novel, and so tell us important properties of how generalizability can change / remain unchanged when a verifier is applied both during learning and at inference time to a pretrained model.

---

> ### Author Response · Authors · 2022-08-02
> **Author response**
>
> Thank you for reviewing our paper. We are happy to hear that you appreciate the importance of the problem setting.
>
>
>
> >  Do you have intuitive explanations for each of the results that could enhance the clarity of the paper?
>
> Thank you for an important suggestion. Theorem 5.1 considers the case that the generalization error of learned hypothesis $\hat{h}$ is  small, i.e, $L_\mathcal{D}(\hat{h}) < \epsilon$ . In such a case,   $L_\mathcal{D}(\hat{h}_c)$ is close to the error of ground truth $f_c$,  and thus it is the smallest among $\mathcal{H}_c$. On the other hand,  Theorem 5.2 considers the case $L_\mathcal{D}(\hat{h}) < \epsilon$   is not satisfied. In such a case, we cannot apply the result of Theorem 5.1.
>
>
>
> For results in Sect.6  (LTV),  adding requirements corresponds to imposing some biases on hypotheses. Therefore, adding $c$ does not increase the complexity of $\mathcal{H}$  would be reasonable, although there are some unintuitive points on proving them (Please see the response to PVsA).
>
>
>
>
>
> > Are there practical settings where the Rademacher complexity based error bounds are known to be meaningful/not terribly loose, and also in which a verifier is useful?
>
>  [Li+ 2017] would be a typical case where the Rademacher complexity-based error bounds work well in practice. This paper reports that using the local Rademacher complexity as a regularizer term in the loss function can improve the classification results with multi-class kernel learning algorithms. Our results suggest that if we use the CV for the problem, we can still use the regularizer term based on the local Rademacher complexity of the original hypothesis class.
>
>
>
> [Li+ 2017] Jian Li, Yong Liu, Rong Yin, Hua Zhang, Lizhong Ding, and Weiping Wang. Multi-class learning: From theory to algorithm, NeurIPS 2018.
>
>
>
>
>
> > You show that Rademacher complexity based error bounds will be not larger than those of the original model when a CV is added in. How does this then allow you to conclude that the CV doesn't hurt the model's generalizability (line 15)? For example in modern ML systems my understanding is that typical generalization performances can be *well* within these bounds, and hence preserving the bounds does not directly entail preserving generalization.
>
>
>
> Thank you for an important suggestion. We show only generalization bounds, and we cannot assess the model's generalization ability. We have removed the last sentence.
>
>
>
>
>
> Thank you for the comments on the crality. We will update the paper following comments.
>
> > Line 185, it should say "sample" rather than "example", yes?
>
> Thank you for the suggestion. We have fixed it.
>
>
>
> > Line 191: The notation f and fc is only introduced in the Appendix, and so seems out of scope here
>
> Thank you for the suggestion. We've added a definition of $f$ and $f_c$ in the main part.
>
>
>
>
>
> >  Line 275: What does f refer to in the definition of empirical local Rademacher complexity?
>
> We've modified the definition of the empirical local Rademacher complexity to $R_S(\mathcal{G}; r) = R_{S}\left(\{a g : a \in [0, 1],  g \in \mathcal{G}, \mathbb{E}[(ag)^2] \leq r \}\right)$.  This modification does not change the theoretical results in this section. Thank you for the suggestion.

---

### Official Review · Reviewer_uBfe · 2022-07-15

**Rating:** 7
**Confidence:** 4
**Soundness:** 3 good
**Presentation:** 3 good
**Contribution:** 3 good

**Summary:**

This paper considers the problem of introducing a concurrent verifier either in the learning or inference phase in the setting of PAC learning.  A concurrent verifier checks if the predicted output satisfies a given requirement (represented as a constraint).  If so, it doesn't change the predicted output.  Otherwise, it forces the output to take a value that satisfies the requirement.  The authors consider the generalization error of learning with concurrent verifiers in the inference and in the learning phase, and show that (a) under the realizability assumption, learning with concurrent verifier in the inference phase doesn't suffer from too much generalization error; (b) for learning with concurrent verifier in the learning phase, the authors use the Rademacher complexity based generalization error bounds and show that these don't increase in the presence of concurrent verifier in the learning phase.  The authors also provide analysis of the running time overhead of using a concurrent verifier in the different settings.  They also describe a structured prediction setting where their theory is applicable.  This is a theoretical paper with no experimental results.

**Questions:**

Questions:
1. Section 5: In the ITV setting, what happens with non 0-1 loss
functions? It doesn't look like the proof of Theorem 5.1 will go
through in this case in general.

2. Line 194 and 199-200: This is a bit mis-leading.  You have not
mentioned the role of the loss function being a 0-1 loss function at
all.  Not only must D be realizable with H, but we also require the
loss function to have special properties.
Shouldn't you change the presentation here?

3. Shouldn't the statement of Theorem 5.2 also include "Suppose
hypothesis class H is PAC-learnable"?

4. Lines 274-275: What is "f" in statement of Definition 6.2?

5. The analysis of running time at the end of Section 6.3 is really
sketchy and doesn't say much concretely.  Why can't this be explained
better?


Some typos and readability hurdles:
Line 156: "if we all a machine": sounds incorrect

Line 191: The notation f_c is introduced suddenly here without any
previous reference in the main paper.  This is inherited from Appendix
A, so unless somebody reads the appendix, they will be confused.

Line 196: Theorem 5.2 -- the statement of the theorem sounds strange.
We are talking about a specific distribution D that is not realizable
with hypothesis class H.  So why do you say "Then, there exists D, ..."

Line 223: "generation error" --> "generalization error"

Theorem 6.3 and the remainder of Section 6.2 are written in a way that
is hard to read and understand.  This part should certainly be
re-written.

Label of Fig. 2(b): Shouldn't we have h_{f_1}(x, y_1, y_2) instead
of h_{f_1}(x, y_1, y_3)?



**Limitations:**

The authors have discussed the limitations of their approach to some extent.  There is no potential negative societal impact of their work.

**Strengths And Weaknesses:**

Strengths:
1. Very relevant problem
2. The results are good and show when the use of concurrent verifiers is beneficial.
3. The results expand our current understanding of learning/predicting with concurrent verifiers
4. The presentation introduces the relevant theory and definitions, making it easier to read.


Weaknesses:
1. The paper is very dense, particularly the statement of the main technical results.  Shifting the proofs to the appendix but not providing any intuition behind the proofs makes it hard to understand the intuition behind the main theorems.
2. The running time complexity analysis for using the concurrent verifier is a bit sketchy.  This is particularly so in Sec 6.3
3. The math is often hard to follow for the general reader.  The paper is unnecessarily notationally heavy at places.

---

> ### Author Response · Authors · 2022-08-02
> **Author response**
>
> Thank you for reviewing our paper. We also thank you for many detailed comments and suggestions.
>
>
>
> >  The paper is very dense, particularly the statement of the main technical results. Shifting the proofs to the Appendix but not providing any intuition behind the proofs makes it hard to understand the intuition behind the main theorems.
>
> Thank you for the comment. We've added some explanations of the main technical results. Please check the attached file. We also show the intuitions behind these theorems in response to reviewer TPUk.
>
>
>
>
>
> >  Section 5: In the ITV setting, what happens with non 0-1 loss functions? It doesn't look like the proof of Theorem 5.1 will go through in this case in general.
>
> Thank you for an interesting question. The current proof is specific for the 0-1 loss function. It is unclear whether 5.1 holds with non 0-1 loss functions.
>
>
>
>
>
> > Line 194 and 199-200: This is a bit mis-leading. You have not mentioned the role of the loss function being a 0-1 loss function at all. Not only must D be realizable with H, but we also require the loss function to have special properties. Shouldn't you change the presentation here?
>
> Thank you for the suggestion. Although we have mentioned the loss function is 0-1 at line 189, it would not be easy to read. We've modified the statement of Theorem 5.2 to mention the loss function. Please check the attached file.
>
>
>
>
>
> >  Shouldn't the statement of Theorem 5.2 also include "Suppose hypothesis class H is PAC-learnable"?
>
> Here we consider the case $\mathcal{D}$  is not realizable and thus $\mathcal{H}$ is **not PAC-learnable**. We've modified the presentation to emphasize the premise condition of Theorem 5.2. Please check the attached file.
>
>
>
>
>
> > Lines 274-275: What is "f" in statement of Definition 6.2?
>
> We've modified the definition of the empirical local Rademacher complexity to $R_S(\mathcal{G}; r) = R_{S}\left(\{a g : a \in [0, 1],  g \in \mathcal{G}, \mathbb{E}[(ag)^2] \leq r \}\right)$. This modification does not change the theoretical results in this section. Thank you for the suggestion.
>
>
>
>
>
> >  The analysis of running time at the end of Section 6.3 is really sketchy and doesn't say much concretely. Why can't this be explained better?
>
> Thank you for the suggestion. We've modified the presentation here as follows:
>
>
>
> We analyze the overhead in running time for evaluating loss function $L_{S, \rho}^{\mathrm{add}}(h_c)$ and $L_{S, \rho}^{\mathrm{mult}}(h_c)$ for hypothesis $h_c$. Unlike the multi-class classification case, the number of queries and the overhead in running time for the loss evaluation when we use a CV depends on the model and the type of requirements for structured prediction. This result is consistent with the literature reporting that for the structured prediction task, it is known that originally
> tractable optimization problems can be intractable if we put additional constraints [21].
>
>
>
>
>
> We also thank you for the comments on typos and readability issues. We will correct them all.
>
> > Some typos and readability hurdles: Line 156: "if we all a machine": sounds incorrect
>
> We've modified it to "if we allow a machine".
>
>
>
> > Line 196: Theorem 5.2 -- the statement of the theorem sounds strange. We are talking about a specific distribution D that is not realizable with hypothesis class H. So why do you say "Then, there exists D, ..."
>
> Thank you for the suggestion. We've modified the first and the second sentences of Theorem 5.2. Please check the attached file.
>
>
>
>
>
> > Theorem 6.3 and the remainder of Section 6.2 are written in a way that is hard to read and understand. This part should certainly be re-written.
>
> We agree that this part is a bit complex. This is because we have to introduce many concepts to state Theorem 6.2 in a self-contained style. We will add more explanations to make this part easier to read in the camera-ready version.
>
>
>
>
>
> > Label of Fig. 2(b): Shouldn't we have h_{f_1}(x, y_1, y_2) instead of h_{f_1}(x, y_1, y_3)?
>
> Thank you for the suggestion. $h_{f_1}(x, y_1, y_2)$ is correct. We have fixed it.

---

### Official Review · Reviewer_PVsA · 2022-07-21

**Rating:** 6
**Confidence:** 2
**Soundness:** 3 good
**Presentation:** 2 fair
**Contribution:** 3 good

**Summary:**

The paper proposes a monitor that observes the input and output of a machine learning model and checks if they satisfy requirements. Two scenarios are investigated - first when the monitor is only present at inference time and the second when the requirement is known at training time.

**Questions:**

- What is fc in Line 191/192? A symbol table in the paper will make it more tractable specially because the theorem and proof are split between the main paper and the appendix.

- Introducing the concurrent verifier can actually lead to increased generalization error based on equation 187. The surprising (and useful) result would upper bound L_D(Hc) instead of lower bound it.

- Looking at the proof of Theorem 5.2, it appears we need to constrain c in some way and the degenerate choice of c is why we get negative result in Theorem 5.2. Can authors help identify a practical scenario or give some intuition where such a c would be defined over input/output?





**Limitations:**

There are no potential negative societal impact of the work to the best of this reviewer's understanding.

**Strengths And Weaknesses:**

+ If a hypothesis class H is PAC learning, then using the verifier at inference stage can give a hypothesis with a guarantee in its generalization error.

+ When requirements are known in training phase, then the input/output requirements do not increase the Rademacher complexity.

- The proofs in appendix are dense and difficult to follow while the theorem statements themselves appear to be rather unsurprising - the clarification questions will help the reviewer better understand the significance of this work.

- The Rademacher bound appears to be straightforward adaption of standard result. It is not clear why one would expect Hc to have higher complexity than H.

---

> ### Author Response · Authors · 2022-08-02
> **Author response**
>
> Thank you for reviewing our paper. Your detailed comments help improve the manuscript.
>
>
>
> > The Rademacher bound appears to be straightforward adaption of standard result. It is not clear why one would expect Hc to have higher complexity than H.
>
> As we have mentioned in lines 259-261, it seems natural to consider that the complexity of $\mathcal{H}_c $ is not higher than $\mathcal{H}$. However, to prove it is not straightforward since $\mathcal{H}_c \not \subseteq \mathcal{H}$. Moreover, in the structured Rademacher complexity case,  the main result (Theorem 6.4) is interesting since the requirements do not need to follow the factor graph structure (lines 337-340).
>
> On the other hand, we believe that the value of our work lies in introducing new problem settings of learning and inferencing with a verifier and giving error bounds in general cases.
>
>
>
>
>
> > What is fc in Line 191/192? A symbol table in the paper will make it more tractable specially because the theorem and proof are split between the main paper and the Appendix.
>
> $f_c$ is made by modifying $f$  to satisfy requirement $c$, where $f$ is a mapping satisfying $L_\mathcal{D}(f ) = 0$, which appears in the proof of Theorem 5.1. We've added the definition of $f$ in the main body of the paper. We've also added a symbol table to the Appendix. Thank you for the suggestion.
>
>
>
>
>
> > Introducing the concurrent verifier can actually lead to increased generalization error based on equation 187. The surprising (and useful) result would upper bound L_D(Hc) instead of lower bound it.
>
> The equation at line 187 gives an upper bound of generalization error  $L_\mathcal{D}(\hat{h}_c)$, and the equation does not show that introducing a verifier increases generalization error. It would be possible that you might misunderstand the left side of the equation with $L_\mathcal{D}(\hat{h})$. We are sorry for using similar notations.
>
>
>
>
>
> > Looking at the proof of Theorem 5.2, it appears we need to constrain c in some way and the degenerate choice of c is why we get negative result in Theorem 5.2. Can authors help identify a practical scenario or give some intuition where such a c would be defined over input/output?
>
> Intuitively, the c appearing in the proof is such that it knows partially about ground truth $f(x)$ when $x \in \mathcal{X}_0$.   Such $c$ helps to reduce the generalization error, but we might fail to select the best hypothesis $h_c \in \mathcal{H}_c$  when we use $c$ only in the inference time. Such $c$  would appear in natural settings where we want to reduce generalization error by injecting knowledge on ground truth $f$.

---

> > ### Comment · Reviewer_PVsA · 2022-08-10
> > **Thank you.**
> >
> > The primary concerns of the reviewer have been met. I am raising the score.

---

### Author Response · Authors · 2022-08-02
**Updating papers**

Thank you for reviewing our paper. We have updated the paper and the supplementary material to reflect comments from reviewers.

---

### Meta-Review · Area_Chair_Rznx · 2022-08-25

**Recommendation:** Accept
**Confidence:** Certain

**Metareview:**

All reviewers liked the presented approach of using a concurrent verifier in both the learning and inference phase in the PAC setting. The approach also presents theoretical proofs for bounds when using such a verifier. The reviews also provide many details for improving the presentation for both improving the correctness and clarity, which would be great to incorporate in the next version of the paper.

**Award:**

No

---

### Decision · Program_Chairs · 2022-09-14

Accept